# Generative Well-intentioned Networks

**Justin Cosentino, Jun Zhu**[*]
Dept. of Comp. Sci. & Tech., Institute for AI, THBI Lab, BNRist Center,
State Key Lab for Intell. Tech. & Sys., Tsinghua University, Beijing, China
justin@cosentino.io, dcszj@mail.tsinghua.edu.cn

## Abstract

We propose Generative Well-intentioned Networks (GWINs), a novel framework for increasing the accuracy of certainty-based, closed-world classifiers. A conditional generative network recovers the distribution of observations that the classifier labels correctly with high certainty. We introduce a reject option to the classifier during inference, allowing the classifier to reject an observation instance rather than predict an uncertain label. These rejected observations are translated by the generative network to high-certainty representations, which are then relabeled by the classifier. This architecture allows for any certainty-based classifier or rejection function and is not limited to multilayer perceptrons. The capability of this framework is assessed using benchmark classification datasets and shows that GWINs significantly improve the accuracy of uncertain observations.

## 1 Introduction

An essential aspect of any machine learning system is understanding what the model does not know. Despite achieving state-of-the-art performance across a wide array of problem domains, current deep learning techniques do not actually capture model uncertainty. Core settings in which standard deep learning approaches have been deployed, such as medical diagnoses, autonomous vehicles, and critical systems, rely on accurate estimates of uncertainty [16, 10]. Though traditional Bayesian probability theory offers mathematical tools to reason about model uncertainty, such approaches do not scale to the high dimensional feature spaces found in many deep learning tasks. The need for principled uncertainty estimates from deep learning architectures has given rise to the field of Bayesian deep learning (see e.g., [35]) and many deep learning techniques have been interpreted through a Bayesian lens with the development of advanced inference algorithms [36, 39], providing novel methods for obtaining uncertainty estimates from deep learning models [21, 11, 12, 13, 22].

One may be able to measure epistemic uncertainty – uncertainty in model prediction due to the lack of knowledge – using Bayesian neural networks [25, 29], but the question of how to best utilize uncertainty estimates still remains. In this paper, we propose Generative Well-intentioned Networks (GWINs), a novel framework that leverages these uncertainty estimates to increase the generalizability and accuracy of certainty-based classifiers. Rather than make low-certainty predictions, a model can reject an observation to achieve an arbitrarily high accuracy [5]. However, a model that refuses to classify is not particularly useful. Borrowing ideas from the fields of classification with rejection and generative networks, we allow a classifier to reject uncertain observations and then, using a generative network, transform them into representations that the classifier labels correctly with high certainty. Informally, one can view the classifier as "intuition" and the generative network as "critical thinking": given a new observation that we can not quickly reason about with prior knowledge, we apply critical thinking to reformulate the problem by relating it to information we already know to be true. We show that the generative network $G$ is able to recover the distribution of observations

---

[*]Corresponding author.

that classifier $C$ labels correctly with high certainty and that this reformulation process significantly increases classifier accuracy on the rejected observation subset.

The rest of this paper is organized as follows. We introduce the necessary background regarding Generative Adversarial Networks (GANs) and rejection-based classification in Section 2. Our proposed GWIN framework is formally defined in Section 3 and a sample GWIN implementation is detailed in Section 4. We then empirically evaluate the effectiveness of the proposed framework in Section 5. Lastly, we discuss related works in Section 6.

## 2 Preliminaries

### 2.1 Generative Adversarial Networks

Generative Adversarial Networks (GANs) [17] are generative models that make use of an adversarial process between two networks to learn a distribution: a generator network $G$ produces synthetic data given some noise vector $z$ while a discriminator network $D$ discriminates between the generator's output and samples from the true data distribution. The goal of the generator is to produce samples that fool the discriminator. Formally, this adversarial game results in the following minimax objective:

$$\min_{G} \max_{D} \mathop{\mathbb{E}}_{\boldsymbol{x} \sim \mathbb{P}_r} [\log(D(\boldsymbol{x}))] + \mathop{\mathbb{E}}_{\tilde{\boldsymbol{x}} \sim \mathbb{P}_g} [\log(1 - D(\tilde{\boldsymbol{x}}))], \tag{1}$$

where $\mathbb{P}_r$ is the real data distribution and $\mathbb{P}_g$ is the generated distribution implicitly defined by $\boldsymbol{x}' = G(\boldsymbol{z})$. $\boldsymbol{z}$ is a random noise vector sampled from a simple noise distribution $p$, i.e., $\boldsymbol{z} \sim p(z)$. With enough capacity, the discriminator will reach an optimum given $G$ so that $\mathbb{P}_r = \mathbb{P}_g$ [17].

It is well known that GANs suffer from training instability [33], suggesting that the divergences which GANs usually minimize are the cause of such training difficulties [2]. The Wasserstein GAN (WGAN) proposes the use of the Earth-Mover distance to define its objective function:

$$\min_{G} \max_{D \in \mathcal{D}} \mathop{\mathbb{E}}_{\boldsymbol{x} \sim \mathbb{P}_r} [D(\boldsymbol{x})] - \mathop{\mathbb{E}}_{\tilde{\boldsymbol{x}} \sim \mathbb{P}_g} [D(\tilde{\boldsymbol{x}})], \tag{2}$$

where $\mathcal{D}$ is the set of 1-Lipschitz functions. The Wasserstein GAN with gradient penalty (WGAN-GP) [19] further builds on this work, providing a final objective function with desirable properties:

$$\min_{G} \max_{D} \mathop{\mathbb{E}}_{\tilde{\boldsymbol{x}} \sim \mathbb{P}_g} [D(\tilde{\boldsymbol{x}})] - \mathop{\mathbb{E}}_{\boldsymbol{x} \sim \mathbb{P}_r} [D(\boldsymbol{x})] + \lambda \mathop{\mathbb{E}}_{\hat{\boldsymbol{x}} \sim \mathbb{P}_{\hat{\boldsymbol{x}}}} \left[ (\|\nabla_{\hat{\boldsymbol{x}}} D(\hat{\boldsymbol{x}})\|_2 - 1)^2 \right]. \tag{3}$$

Lastly, GANs can be extended to conditional models by conditioning both the discriminator and generator on auxiliary information $\boldsymbol{y}$ [27]. By providing $\boldsymbol{y}$ as additional input to each network, the original GAN objective function presented in Equation (1) becomes:

$$\min_{G} \max_{D} \mathop{\mathbb{E}}_{\boldsymbol{x} \sim \mathbb{P}_r} [\log(D(\boldsymbol{x}, \boldsymbol{y}))] + \mathop{\mathbb{E}}_{\boldsymbol{z} \sim \mathbb{P}_z} [\log(1 - D(G(\boldsymbol{z}, \boldsymbol{y}), \boldsymbol{y}))]. \tag{4}$$

In this work, we build upon a conditional implementation of the WGAN with gradient penalty.

### 2.2 Classification with Reject

Entirely orthogonal to the field of generative networks is the study of classification with rejection. The problem of classification with rejection can be informally defined as giving the classifier the option to reject an observation instance instead of predicting its label. Depending on the setting, the classifier may incur some small cost for rejection, though this cost is typically less than that of a random prediction. The motivation behind rejection-based classification is to avoid misclassification in high risk situations, such as medical diagnoses, when the classifier has low certainty that its prediction will be correct. Early works explored the inherent tradeoff between error rate and rejection rate [4, 5], while more recent works have explored the binary classification setting [37, 3, 6]. We borrow the basic idea of threshold rejection from these works: given some threshold $\tau$, one rejects an observation instance if certainty in correct prediction is less than $\tau$.

Recent work also explored the reject option in the context of deep learning [14, 15]. Though we opt for the simplicity of the thresholded reject option described above, it is worth noting that these methodologies could also be used within the Generative Well-intentioned Network framework.

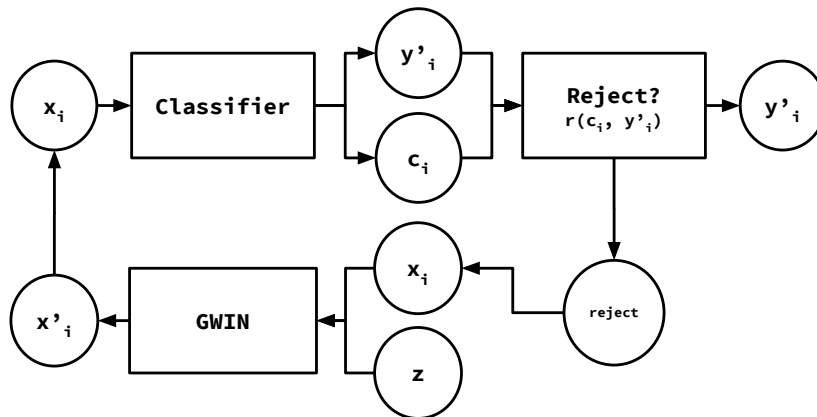

Figure 1: The inference process for some new observation $x_i$. If classifier $C$ labels the input $y_i'$ with certainty $c_i$ and rejects the query, the conditional GWIN translates the given query to the classifier's confident distribution. The transformed query $x_i'$ is then relabeled by the classifier, i.e., $C(G(x_i, z))$. The variable $z$ denotes a random noise vector. The top half of this figure outlines the expected interface of the rejection-based classifier. Aside from requiring the model to emit a certainty metric $c_i$ and label $y_i'$, no strong assumptions are made about the classifier. Since the classifier is fixed during generative training, it need not be a perceptron-based model. The rejection function $r : \{(c, y')\} \rightarrow \{\text{reject}, y'\}$ determines if the given observation is rejected or labeled.

## 3   Generative Well-intentioned Network Framework

We propose a novel framework that leverages uncertainty estimates and generative networks to increase the accuracy of certainty-based models during inference. The framework consists of three core components:

1. A pretrained, certainty-based classifier $C$ that emits a prediction $y_i'$ with certainty $c_i$ when labeling a new observation $x_i$, i.e., $(y_i', c_i) = C(x_i)$

2. A rejection function $r : \{(c, y')\} \rightarrow \{\text{reject}, y'\}$ that allows the classifier to reject an uncertain instance rather than predicting its label

3. A conditional generative network $G$ that transforms an observation $x_i$ and noise vector $z$ to a new representation $x_i'$, i.e., $x_i' = G(x_i, z)$

A key feature of this framework is that it can be used together with any certainty-based classifier and does not modify the classifier structure at any point during the generative training process. Assuming that the classifier and rejection function provide the interface illustrated in Figure 1, any classifier or rejection function can be used within this framework.

Given this fixed, certainty-based classifier $C$, the conditional GWIN $G$ learns distribution $\mathbb{P}_c$, where $\mathbb{P}_c$ represents the distribution of observations from the original data distribution $\mathbb{P}_r$ that $C$ labels correctly with high certainty. The goal of $G$ is to generate a new observation $x' \sim \mathbb{P}_c$ from $(x, y) \sim \mathbb{P}_r$ that the classifier will label as ground truth $y$ with high certainty. During inference, the classifier can choose to reject observation $x$ if uncertain that it will label $x$ correctly. This observation is then passed to $G$, along with a noise vector $z$, to generate a transformed sample for reclassification. The inference process is illustrated in Figure 1 and examples of the transformation process using a Wasserstein GWIN are shown in Figure 2.

Similarly to the classifier and the rejection function, we do not place any strong restrictions on the generative framework. We propose a Wasserstein GWIN in Section 4 as one potential approach. Though the Wasserstein network makes use of adversarial procedure, we refer to these generative networks as "well-intentioned" since they aim to maximize the accuracy and certainty of the provided classifier.

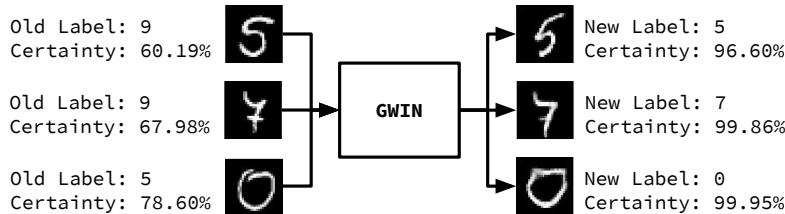

Figure 2: A visual representation of the GWIN transformation using example images from the MNIST Digits dataset. With a certainty threshold of $\tau = 0.8$, the classifier rejects the observations on the left, which would had been labeled incorrectly were the classifier forced to predict. These observations are then transformed into the representations on the right using the Wasserstein GWIN described in Section 4.3. When relabeling the generated images, i.e., $C(G(\boldsymbol{x}, \boldsymbol{z}))$, the classifier labels correctly with high-certainty.

## 4  Wasserstein Generative Well-intentioned Network

We outline a sample GWIN implementation, as defined in Section 3, based on the Wasserstein GAN [2]. We utilize a Bayesian Neural Network classifier and a simple $\tau$-threshold rejection function. Section 5 evaluates this proposed implementation.

### 4.1  Classifier

The GWIN is paired with a Bayesian neural network [29] using a LeNet-5 architecture [23]. A detailed description of the classifier's architecture is in the appendix. The network is implemented using TensorFlow Probability [7], which provides clean abstractions for Bayesian variational inference. The model uses the Flipout estimator [40] to minimize the Kullback-Leibler divergence up to a constant, also known as the negative Evidence Lower Bound (ELBO).

We approximate prediction certainty using Monte Carlo sampling to draw class probabilities from the model. We treat the median prediction of these draws as the certainty metric for each class and the mean prediction value as the prediction score. The class with the highest prediction value and its certainty metric are then provided to the rejection function.

Recall from Section 3 that the GWIN Framework is model-agnostic for certainty-based classifiers. Thus, experiments do not focus on improving the classifier or rejection function, but rather analyze how the GWIN improves accuracy for a fixed classifier. In the appendix, we show that the GWIN still improves classifier performance for a stronger Bayesian neural network.

### 4.2  Rejection Function

We use a simple $\tau$-threshold rejection rule, where $\tau \in [0, 1]$:

$$r(c_i, y_i') = \begin{cases} y_i', & \text{if } c_i \geq \tau \\ \text{reject}, & \text{otherwise.} \end{cases} \tag{5}$$

The choice of $\tau$ is made at time of inference, meaning that this rejection function can be tuned after the generative network has been trained for optimal accuracy. Setting $\tau = 0$ rejects no values and is equivalent to using only the base classifier, while setting $\tau = 1$ rejects all values and is equivalent to preprocessing all input with the GWIN.

### 4.3  Wasserstein GWIN with Gradient Penalty

The Wasserstein GWIN with gradient penalty (WGWIN-GP) is based on the Wasserstein GAN with gradient penalty [19]. The architectures of both the critic and generator closely follow the original WGAN-GP models and a detailed description of these architectures is in the appendix. In this subsection, we detail core modifications to the original model.

**Loss with Transformation Penalty**   The WGWIN-GP introduces a new loss function with a transformation penalty that encourages the conditional generator to produce images that the classifier will label correctly. Given some $(\boldsymbol{x_i}, y_i)$ training observation, the generator should produce $\boldsymbol{x_i'}$ that the classifier labels as $y_i$. This penalty is the loss of the classifier when labeling the transformed observations in the current training batch, denoted $\text{Loss}(C(\boldsymbol{x'}))$. We include a penalty coefficient $\lambda_{Loss}$. All experiments in this paper use $\lambda_{Loss} = 10$, which we found to work well across experiments. Equation (6) shows the loss function for the GWIN:

$$L = \underbrace{\mathop{\mathbb{E}}_{\boldsymbol{x'} \sim \mathbb{P}_g}[D(\boldsymbol{x'}, y)] - \mathop{\mathbb{E}}_{\boldsymbol{x} \sim \mathbb{P}_c}[D(\boldsymbol{x}, y)]}_{\text{WGAN Loss}} + \underbrace{\lambda_{GP} \mathop{\mathbb{E}}_{\hat{\boldsymbol{x}} \sim \mathbb{P}_{\hat{\boldsymbol{x}}}}[(||\nabla_{\hat{\boldsymbol{x}}} D(\hat{\boldsymbol{x}}, y)||_2 - 1)^2]}_{\text{WGAN-GP Penalty}} + \underbrace{\lambda_{Loss} \mathop{\mathbb{E}}_{\boldsymbol{x'} \sim \mathbb{P}_g}[\text{Loss}(C(\boldsymbol{x'}))]}_{\text{Transformation Penalty}}. \quad (6)$$

**Critic Training on Confident Subset**   The WGAN-GP critic is typically trained on both generated data $\boldsymbol{x'} \sim \mathbb{P}_g$ and real data $\boldsymbol{x} \sim \mathbb{P}_r$. However, we want the GWIN to generate images from the classifier's confident distribution. Thus, we prefilter the training data to create a confident distribution $\mathbb{P}_c$ containing all images that the classifier labels correctly with certainty of at least $\tau^*$. The critic is then trained exclusively on samples drawn from $\mathbb{P}_c$ and $\mathbb{P}_g$. Note that $\tau^*$ is not necessarily the same certainty threshold used in the rejection function. We set $\tau^*$ to some arbitrarily high certainty, e.g., 0.95, so that the rejection function can be tuned without needing to retrain the generative model.

Since the WGWIN-GP will encounter observations from $\mathbb{P}_r$ during inference, only the critic samples from $\mathbb{P}_c$. During training, the generator samples from the entire real distribution $\mathbb{P}_r$.

**A Conditional Generative Model**   The WGWIN-GP is trained as a conditional GAN. Conditional generative networks are often class conditioned to generate an example of a specific class, and the same conditioning information is given to both the critic and generator. However, as the WGWIN-GP will not have access to the ground truth label during inference, the generator is conditioned on the entire observation $\boldsymbol{x}$. We want the critic to discriminate between certain and uncertain observations. Since $\boldsymbol{x}$ is not guaranteed to be from $\mathbb{P}_c$, we condition the critic on a one-hot representation of the ground truth label $y$ in an effort to generate images that are representative of the original observation's class. Thus the generator is tasked with translating observations to new images that are from the given class in the confident distribution.

One can achieve conditioning by concatenating the conditional information with the input [27] or with a feature vector at some hidden layer within the network [32, 42]. Though other conditioning methods exists, such as modifying the discriminator's loss function to also maximize the log likelihood of the correct class [30] or projection-based approaches [28], we opted to condition the generator using input-based concatenation and to condition the critic using hidden-layer concatenation for simplicity.

Algorithm1 shows the new WGWIN-GP training algorithm.

# 5   Evaluation

We evaluate the WGWIN-GP using the training procedure outlined in Section 4 and the inference method illustrated in Figure 1. We compare test accuracy of the base Bayesian neural network, denoted *BNN*, the Bayesian neural network with reject, denoted *BNN w/Reject*, and the Bayesian neural network when paired with the WGWIN-GP, denoted *BNN+GWIN*. BNN w/Reject allows the classifier to reject observations without needing to relabel while the BNN+GWIN uses the WGWIN-GP to transform and relabel the rejected subset.

The BNN trained for 30 epochs using a learning rate of 0.001 and batch size of 128. The GWIN trained for 200,000 iterations using the default hyperparameters listed in Algorithm 1. Both the generator and critic used a learning rate of 0.0001 and batch size of 128. We perform inference using various certainty thresholds $\tau \in \{0.10, 0.30, 0.50, 0.70, 0.80, 0.90, 0.95, 0.99\}$. The BNN uses 10 Monte Carlo samples to determine prediction certainty.

Given the non-deterministic nature of both the Bayesian neural network and the generative network, all experimental results are averaged over 10 runs. We trained and evaluated the models using NVIDIA GeForce GTX TITAN X GPUs.

**Algorithm 1:** WGWIN with gradient and transformation penalty. We use default values of $\lambda_{GP} = 10$, $\lambda_{Loss} = 10$, $n_{\text{critic}} = 5$, $\alpha = 0.0001$, $\beta_1 = 0.5$, $\beta_2 = 0.9$, certainty preprocessing threshold $\tau^* = 0.95$ and the fixed classifier $C$ described in Section 4.1.

**Require :** The penalty coefficients $\lambda_{GP}$ and $\lambda_{Loss}$, the number of critic iterations per generator iteration $n_{\text{critic}}$, the batch size $m$, Adam hyperparameters $\alpha, \beta_1, \beta_2$, certainty preprocessing threshold $\tau^*$, and classifier $C$.

**Require :** initial critic parameters $w_0$, initial generator parameters $\theta_0$

1:   Build confident data distribution $\mathbb{P}_c$ from training data $\mathbb{P}_r$ using classifier $C$ and threshold $\tau^*$
2:   **while** $\theta$ *has not converged* **do**
3:      **for** $t = 1, \ldots, n_{critic}$ **do**
4:        **for** $i = 1, \ldots, m$ **do**
5:          Sample confident data $(\boldsymbol{x}, y) \sim \mathbb{P}_c$, latent variable $\boldsymbol{z} \sim p(\boldsymbol{z})$, and a random number $\epsilon \sim U[0, 1]$.
6:          $\boldsymbol{x}' \leftarrow G_\theta(\boldsymbol{x}, \boldsymbol{z})$
7:          $\hat{\boldsymbol{x}} \leftarrow \epsilon\boldsymbol{x} + (1 - \epsilon)\boldsymbol{x}'$
8:          $L^{(i)} \leftarrow D_w(\boldsymbol{x}', y) - D_w(\boldsymbol{x}, y) + \lambda_{GP}(||\nabla_{\hat{\boldsymbol{x}}}D_w(\hat{\boldsymbol{x}}, y)||_2 - 1)^2$
9:        **end for**
10:        $w \leftarrow \text{Adam}(\nabla_w \frac{1}{m}\sum_{i=1}^m L^{(i)}, w, \alpha, \beta_1, \beta_2)$
11:      **end for**
12:      Sample a batch of training data $\{(\boldsymbol{x}, y)^{(i)}\}_{i=1}^m \sim \mathbb{P}_r$ and latent variables $\{\boldsymbol{z}^{(i)}\}_{i=1}^m \sim p(z)$
13:      $\theta \leftarrow \text{Adam}(\nabla_\theta \frac{1}{m}\sum_{i=1}^m -D_w(G_\theta(\boldsymbol{x}, \boldsymbol{z}), y) + \lambda_{Loss}(\text{Loss}(C(G_\theta(\boldsymbol{x}, \boldsymbol{z})))), \theta, \alpha, \beta_1, \beta_2)$
14: **end while**

## 5.1   Datasets

We use two different datasets in our experiments: the MNIST handwritten digits [23] dataset and the Fashion-MNIST clothing dataset [41]. Both datasets consist of 60,000 training images and 10,000 test images. We further split both training sets into a 50,000 image training set and 10,000 image validation set. Each example is a 28x28x1 grayscale image associated with a label from one of ten classes. Images are preprocessed by normalizing grayscale values to $[0, 1]$.

Building the certain distribution $\mathbb{P}_c$ filters each dataset a varying amount. The average size of the high certainty training dataset is 47,948 for MNIST Digits and 31,760 for MNIST Fashion.

## 5.2   Results

Figure 3 and Figure 4 illustrate the mean accuracy for varying certainty rejection thresholds on each dataset while Table 1 and Table 2 present exact accuracy values on the rejected subset. At every certainty threshold, the GWIN+BNN outperforms the BNN on uncertain observations by up to 35% on MNIST Digits and 20% on MNIST Fashion. As the certainty threshold increases, we see the size of the rejected subset increase and the relative gains from the GWIN transformation decrease. However, this is expected as we begin to reject observations that the BNN already labels correctly with higher certainty. Figure 5 shows the change in certainty of the ground truth label at varying certainty rejection thresholds. Though the GWIN increases certainty in the ground truth label in the majority of observations, it is possible for the GWIN to map an observation to a lower-certainty representation. This suggests that one must carefully tune the rejection function and certainty metrics to minimize the number of correct instances that are mistranslated.

# 6   Related Work

Classifiers and inference networks have been paired with generative adversarial networks in the past, but the goal of these models has been to either learn a mapping from data to latent representations or improve class-conditional generation [8, 9, 24]. Though GWINs also contain an additional classification network, the objective of the generative network is not solely image synthesis or

Table 1: Test set accuracy for MNIST Digits on rejected observations using GWIN transformation for the given certainty threshold $\tau$. *BNN* and *BNN+GWIN* denote accuracy for the rejected subset using only the BNN and the BNN with GWIN reformulation, respectively. With no rejections ($\tau = 0$), the BNN had an accuracy of 98.0%. *Overall Acc.* $\Delta$ is the change in accuracy while *% Error* $\Delta$ denotes the percent change in error rate for the entire subset when the GWIN is applied to rejected queries. All results are presented as the mean over 10 runs.

| $\tau$ | % Reject | BNN Acc. | BNN+GWIN Acc. | Rejected Acc. $\Delta$ | Overall Acc. $\Delta$ | % Error $\Delta$ |
|---|---|---|---|---|---|---|
| 0.50 | 0.39 | $40.23 \pm 8.51$ | $75.59 \pm 4.22$ | $35.36 \pm 8.66$ | $0.14 \pm 0.04$ | $-6.98 \pm 2.08$ |
| 0.70 | 1.83 | $54.48 \pm 2.21$ | $85.07 \pm 2.63$ | $30.59 \pm 2.64$ | $0.56 \pm 0.06$ | $-27.55 \pm 2.66$ |
| 0.80 | 2.74 | $58.91 \pm 1.49$ | $86.30 \pm 1.85$ | $27.39 \pm 2.03$ | $0.75 \pm 0.06$ | $-36.36 \pm 1.93$ |
| 0.90 | 4.39 | $68.79 \pm 2.38$ | $86.95 \pm 0.97$ | $18.16 \pm 2.55$ | $0.80 \pm 0.13$ | $-40.26 \pm 4.19$ |
| 0.95 | 6.04 | $73.48 \pm 1.66$ | $89.34 \pm 0.85$ | $15.86 \pm 2.07$ | $0.96 \pm 0.13$ | $-47.45 \pm 4.09$ |
| 0.99 | 11.00 | $83.54 \pm 0.88$ | $92.55 \pm 0.49$ | $9.02 \pm 0.94$ | $0.99 \pm 0.10$ | $-49.45 \pm 3.16$ |

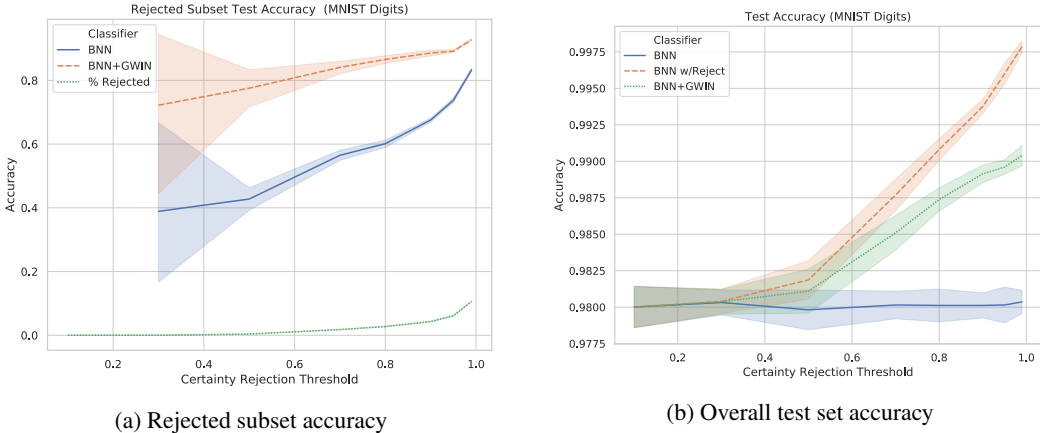

(a) Rejected subset accuracy          (b) Overall test set accuracy

Figure 3: Test set accuracy for MNIST Digits using GWIN transformation for the given certainty threshold $\tau$. Figure 3a shows *BNN* and *BNN+GWIN* accuracy on the rejected subset. *% Reject* represents the percent of the 10,000 observations rejected by the classifier for the current certainty threshold. Figure 3b shows the accuracy of the *BNN* and *BNN+GWIN* on the entire test set. All results are presented as the mean over 10 runs and error bars show standard deviation.

uncovering latent factors, but rather is to reprocess observations in order to increase the classifier's generalizability and accuracy.

To the best of our knowledge, Defense-GAN is the only other instance of pairing a GAN with a classification network to increase performance during inference [34]. Defense-GAN serves as a defense against adversarial examples by using a GAN to "denoise" perturbed images prior to classification. A WGAN is first trained to capture the unperturbed training distribution. Before to labeling a new observation $x$, the image is projected onto the range of the generator by minimizing the reconstruction error,

$$\min_{z} ||G(z) - x||_2^2,$$

using $L$ steps of gradient descent for $R$ different samples of $z$.

Though both Defense-GAN and GWINs use WGAN-based implementations to improve classifier inference, there are a number of differences between these two generative models that stem from the differences in the problems the attempt to solve:

- Defense-GAN aims to denoise adversarial examples by projecting images back to the real data set while minimizing reconstruction loss. However, this assumes that there exists a denoised equivalent of each observation in the real dataset. GWINs, on the other hand, use a conditional WGAN in order to create high-certainty representations of the same class as the original observation.

Table 2: Test set accuracy for MNIST fashion on rejected observations using GWIN transformation for the given certainty threshold $\tau$. *BNN* and *BNN+GWIN* denote accuracy for the rejected subset using only the BNN and the BNN with GWIN reformulation, respectively. With no rejections ($\tau = 0$), the BNN had an accuracy of $87.4\%$. *Overall Acc.* $\Delta$ denotes the change in accuracy while *% Error* $\Delta$ denotes the percent change in error rate for the entire subset when the GWIN is applied to rejected queries. All results are presented as the mean over 10 runs.

| $\tau$ | % Reject | BNN Acc. | BNN+GWIN Acc. | Rejected Acc. $\Delta$ | Overall Acc. $\Delta$ | % Error $\Delta$ |
|---|---|---|---|---|---|---|
| 0.50 | 4.18 | $40.52 \pm 2.36$ | $59.43 \pm 2.30$ | $18.91 \pm 3.61$ | $0.79 \pm 0.17$ | $-6.22 \pm 1.24$ |
| 0.70 | 15.25 | $52.08 \pm 1.55$ | $66.95 \pm 0.67$ | $14.87 \pm 1.78$ | $2.27 \pm 0.30$ | $-18.08 \pm 1.98$ |
| 0.80 | 21.21 | $57.87 \pm 0.89$ | $69.16 \pm 0.47$ | $11.29 \pm 0.87$ | $2.39 \pm 0.19$ | $-19.25 \pm 1.32$ |
| 0.90 | 30.29 | $64.14 \pm 0.66$ | $73.18 \pm 0.73$ | $9.04 \pm 0.83$ | $2.74 \pm 0.29$ | $-21.63 \pm 1.85$ |
| 0.95 | 37.30 | $68.93 \pm 0.49$ | $76.06 \pm 0.43$ | $7.14 \pm 0.61$ | $2.66 \pm 0.25$ | $-21.15 \pm 1.61$ |
| 0.99 | 51.97 | $76.55 \pm 0.30$ | $81.34 \pm 0.26$ | $4.79 \pm 0.34$ | $2.49 \pm 0.19$ | $-19.94 \pm 1.30$ |

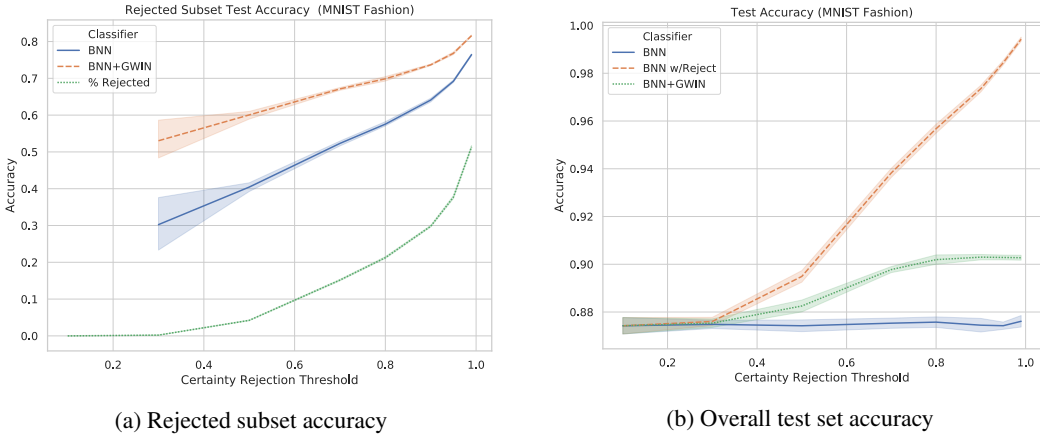

(a) Rejected subset accuracy  (b) Overall test set accuracy

Figure 4: Test set accuracy for MNIST Fashion using GWIN transformation for the given certainty threshold $\tau$. Figure 4a shows *BNN* and *BNN+GWIN* accuracy on the rejected subset. *% Reject* represents the percent of the 10,000 observations rejected by the classifier for the current certainty threshold. Figure 4b shows the accuracy of the *BNN* and *BNN+GWIN* on the entire test set. All results are presented as the mean over 10 runs and error bars show standard deviation.

- Defense-GAN preprocesses all input to the classifier, incurring the cost of the $R \times L$ generations to label each observation. GWINs only transform rejected observations and require at most a single pass through the generator. We include notes on transformation latency for MNIST experiments in the appendix.

- GWINs make stronger assumptions about the classifier than Defense-GAN, requiring a certainty metric and reject function, but can be used for any classification task and are not limited to adversarial robustness.

- GWINs use the fixed classifier during training, while Defense-GAN is trained independently.

The novel contribution of GWINs is using the generative network to learn $\mathbb{P}_c$ of a certainty-based classifier. The WGWIN-GP is just one possible implementation of this idea; though Defense-GAN is structured differently to address adversarial examples, one could imagine a similar method being applied as a new GWIN implementation. We leave this for future work.

Similarly to both DefenseGAN and GWINs, MagNet [26] is a framework that contains a detector network that learns to differentiate between normal and adversarial examples and a reformer network that moves adversarial examples towards the manifold of normal examples in order to protect against adversarial examples with small perturbations. Though this seems to be the second closest model to GWINs, MagNet relies on auto-encoders and also focuses on increasing a model's robustness to adversarial examples rather than making use of classifier certainty to label novel examples from the normal manifold.

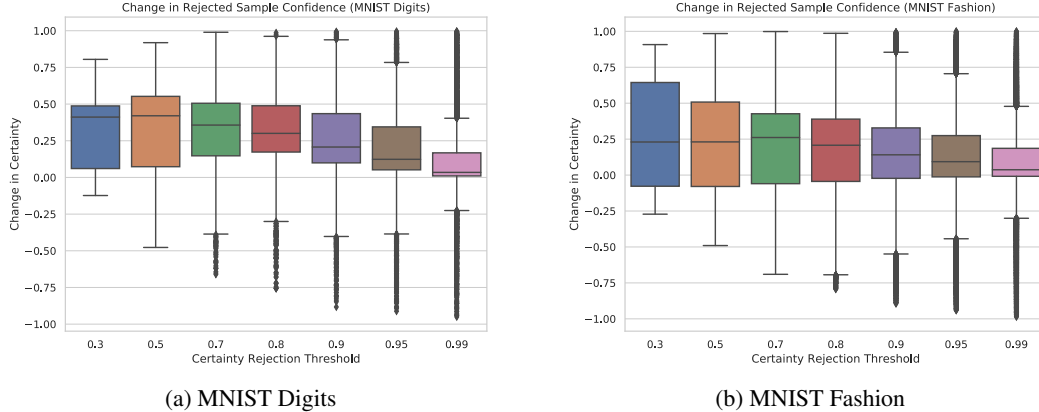

(a) MNIST Digits             (b) MNIST Fashion

Figure 5: Change in rejected sample certainty of the ground truth label for varying certainty rejection thresholds $\tau$. Outliers are those values that fall outside of 1.5IQR and are denoted with diamonds.

Other common strategies for denoising adversarial examples do not translate well to the uncertainty-rejection paradigm. Network distillation [31] trains a classifier such that it is nearly impossible to generate adversarial examples using gradient-based attacks. However, novel observations that might make a classifier uncertain in its predictions are not necessarily generated in an adversarial manner and thus we have no need to mask the network's gradients. Adversarial training [18] is specific to the attack generating the adversarial examples and does not necessarily generalize well to other attacks. Methods that generate additional training data, similarly to hallucination methods in the few-shot learning domain [1, 20, 38], aim to increase the robustness of a classifier *during training* by generating out-of-distribution training data while our method assumes a *fixed, pretrained* classifier and uses generative methods to translate novel, out-of-distribution examples to the confident distribution *during inference*. Since the GWIN framework learns representations that the classifier labels correctly with high confidence, these generative denoising methods can easily be paired with our framework: a classifier is trained using the aforementioned techniques and the GWIN is then used to transform any novel examples that the new classifier is not entirely robust to. Similarly to DefenseGAN and MagNet, the flexibility and additive nature of our frameworks means that we can easily build atop these existing denoising methodologies. Since noise only represents a subset of out-of-distribution observations, we cannot rely entirely on denoising techniques to address classifier robustness. GWINs take a step towards a generalizable, principled framework for "rethinking" uncertain examples and leveraging classifier uncertainty.

# 7 Conclusion

In this work, we outlined Generative Well-intentioned Networks (GWINs), a novel framework leveraging uncertainty and generative networks to increase classifier accuracy. We proposed a high level architecture making use of certainty-based classifiers, a rejection function, and a generative network. We defined a baseline implementation, the Wasserstein GWIN with gradient penalty (WGWIN-GP), and empirically showed that the WGWIN-GP outperforms the base Bayesian neural network at all certainty thresholds. This paper has demonstrated the viability of the GWIN framework and we hope that our work leads to further study of the use of generative networks to aid classifier inference.

# Acknowledgements

This work was supported by the National Key Research and Development Program of China (No. 2017YFA0700904), NSFC Projects (Nos. 61620106010, 61621136008, 61571261), Beijing NSF Project (No. L172037), Beijing Academy of Artificial Intelligence (BAAI), Tiangong Institute for Intelligent Computing, the JP Morgan Faculty Research Program, and the NVIDIA NVAIL Program with GPU/DGX Acceleration.

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
