[Supplementary Material]

# Supplementary Materials for Generative Well-intentioned Networks

**Justin Cosentino, Jun Zhu**[*]
Dept. of Comp. Sci. & Tech., Institute for AI, THBI Lab, BNRist Center,
State Key Lab for Intell. Tech. & Sys., Tsinghua University, Beijing, China
`justin@cosentino.io, dcszj@mail.tsinghua.edu.cn`

## S1  Network Architectures

The LeNet-5 Bayesian neural network model closely follows the standard LeNet-5 architecture, replacing convolutional and dense layers with probabilistic layers from TensorFlow Probability [2]. The model uses the Flipout estimator [8] to minimize the Kullback-Leibler divergence up to a constant. Table 1 contains a detailed description of the network's architecture.

The architectures of the WGWIN-GP critic and generator closely follow those described in the WGAN-GP paper [1]. We add conditional inputs to both networks. The critic is conditioned on the one-hot representation of the class label, which is depth-wise concatenated to both the input and hidden layers of the model [7, 9]. Table 2 details the critic's architecture. The generator is conditioned on the rejected input image, which is flattened and concatenated to the random noise vector [6]. Table 3 details the generator's architecture.

## S2  Improved Bayesian Neural Network Baseline

We use the simple LeNet-5 BNN as a proof of concept for the Generative Well-intentioned Network framework. In order to assess the impact of a GWIN when paired with a stronger classifier, we also repeat experiments using an improved BNN architecture. We see that the GWIN still has a positive, though less pronounced, impact on the rejected subset.

### S2.1  Network Architecture

Table 4 details the Improved BNN (IBNN) baseline's architecture.

### S2.2  Results

Figure 1 and Figure 2 illustrate the mean accuracy for varying certainty rejection thresholds on each dataset while Table 5 and Table 6 present exact accuracy values on the rejected subset. At most certainty thresholds, the GWIN+Improved BNN outperforms the Improved BNN on uncertain observations. As the certainty threshold increases, we see the size of the rejected subset increase and the relative gains from the GWIN transformation decrease. However, this is expected as we begin to reject observations that the Improved BNN already labels correctly with higher certainty. Figure 3 shows the change in certainty of the ground truth label at varying certainty rejection thresholds. Though the GWIN typically increases certainty in the ground truth label in the majority of observations, it is possible for the GWIN to map an observation to a lower-certainty representation. This suggests that one must carefully tune the rejection function and certainty metrics to minimize the number of correct instances that are mistranslated.

---

[*]Corresponding author.

Table 1: Bayesian LeNet-5 model architecture [5] used as a baseline classifier. "Flipout" denotes TensorFlow Probability [2] layers using a Flipout estimator [8].

| $C(\boldsymbol{x})$ | | | | | | |
|---|---|---|---|---|---|---|
| Operation | Kernel | Strides | Padding | Filters | Output Shape | Nonlinearity |
| Conv2D (Flipout) | 5×5 | 1×1 | same | 6 | 28×28×6 | ReLU |
| MaxPooling2D | 2×2 | 2×2 | same | - | 14×14×6 | - |
| Conv2D (Flipout) | 5×5 | 1×1 | same | 16 | 14×14×16 | ReLU |
| MaxPooling2D | 2×2 | 2×2 | same | - | 7×7×16 | - |
| Conv2D (Flipout) | 5×5 | 1×1 | same | 120 | 7×7×120 | ReLU |
| Flatten | - | - | - | - | 5880 | - |
| Dense (Flipout) | - | - | - | - | 84 | ReLU |
| Dense (Flipout) | - | - | - | - | 10 | - |

Table 2: Conditional WGAN-GP-based critic architecture [1]. "Concatenation" denotes depth-wise concatenation of the given one-hot label to the input image as conditional input [7, 9].

| $D(\boldsymbol{x}, y)$ | | | | | | |
|---|---|---|---|---|---|---|
| Operation | Kernel | Strides | Padding | Filters | Output Shape | Nonlinearity |
| Concatenation | - | - | - | - | 28×28×11 | - |
| Conv2D | 5×5 | 2×2 | same | 64 | 14×14×64 | Leaky ReLU |
| Concatenation | - | - | - | - | 14×14×74 | - |
| Conv2D | 5×5 | 2×2 | same | 128 | 7×7×128 | Leaky ReLU |
| Concatenation | - | - | - | - | 7×7×138 | - |
| Conv2D | 5×5 | 2×2 | same | 256 | 4×4×256 | Leaky ReLU |
| Concatenation | - | - | - | - | 4×4×266 | - |
| Flatten | - | - | - | - | 4256 | - |
| Dense | - | - | - | - | 1 | - |

Table 3: Conditional WGAN-GP-based generator architecture [1]. "Concatenation" denotes concatenation of the given flattened image to the input noise as conditional input [6].

| $G(\boldsymbol{x}, \boldsymbol{z})$ | | | | | |
|---|---|---|---|---|---|
| Operation | Kernel | Strides | Padding | Output Shape | Nonlinearity |
| Concatenation | - | - | - | 884 | - |
| Dense | - | - | - | 4096 | ReLU |
| Reshape | - | - | - | 4×4×256 | ReLU |
| Conv2D Transpose | 5×5 | 2×2 | same | 8×8×128 | ReLU |
| Cropping2D | - | - | - | 7×7×128 | - |
| Conv2D Transpose | 5×5 | 2×2 | same | 14×14×64 | ReLU |
| Conv2D Transpose | 5×5 | 2×2 | same | 28×28×1 | Sigmoid |

Table 4: Improved Bayesian Neural Network model architecture used as a baseline classifier. "Flipout" denotes TensorFlow Probability [2] layers using a Flipout estimator [8]. "BN?" and "Dropout" denote whether or not batch norm or dropout were applied after the given layer, respectively.

| | | | | $C(\boldsymbol{x})$ | | | | |
|---|---|---|---|---|---|---|---|---|
| Operation | Kernel | Strides | Padding | Filters | Output Shape | Nonlinearity | BN? | Dropout |
| Conv2D (Flipout) | 3×3 | 1×1 | valid | 32 | 26×26×32 | ReLU | × | - |
| Conv2D (Flipout) | 3×3 | 1×1 | valid | 32 | 24×24×32 | ReLU | × | - |
| Conv2D (Flipout) | 5×5 | 2×2 | same | 32 | 12×12×32 | ReLU | × | 0.4 |
| Conv2D (Flipout) | 3×3 | 1×1 | valid | 64 | 10×10×64 | ReLU | × | - |
| Conv2D (Flipout) | 3×3 | 1×1 | valid | 64 | 8×8×64 | ReLU | × | - |
| Conv2D (Flipout) | 5×5 | 2×2 | same | 64 | 4×4×64 | ReLU | × | 0.4 |
| Flatten | - | - | - | - | 1024 | - | - | - |
| Dense (Flipout) | - | - | - | - | 128 | ReLU | × | 0.4 |
| Dense (Flipout) | - | - | - | - | 10 | - | - | - |

Table 5: Test set accuracy for MNIST Digits on rejected observations using GWIN transformation for the given certainty threshold $\tau$. *BNN* and *BNN+GWIN* denote accuracy for the rejected subset using only the Improved BNN and the Improved BNN with GWIN reformulation, respectively. With no rejections ($\tau = 0$), the Improved BNN had an accuracy of 99.1%. *Overall Acc.* $\Delta$ is the change in accuracy while *% Error* $\Delta$ denotes the percent change in error rate for the entire subset when the GWIN is applied to rejected queries. All results are presented as the mean over 10 runs.

| $\tau$ | % Reject | IBNN Acc. | IBNN+GWIN Acc. | Rejected Acc. $\Delta$ | Overall Acc. $\Delta$ | % Error $\Delta$ |
|---|---|---|---|---|---|---|
| 0.70 | 0.25 | $43.88 \pm 7.83$ | $56.38 \pm 10.87$ | $12.50 \pm 14.17$ | $0.03 \pm 0.03$ | $-3.34 \pm 3.52$ |
| 0.80 | 0.39 | $49.32 \pm 5.74$ | $58.33 \pm 6.14$ | $9.01 \pm 7.81$ | $0.04 \pm 0.03$ | $-3.74 \pm 3.11$ |
| 0.90 | 0.59 | $52.05 \pm 7.99$ | $60.41 \pm 6.10$ | $8.36 \pm 8.58$ | $0.05 \pm 0.05$ | $-5.21 \pm 5.42$ |
| 0.95 | 0.79 | $53.92 \pm 5.42$ | $61.50 \pm 5.04$ | $7.58 \pm 6.97$ | $0.06 \pm 0.06$ | $-5.98 \pm 5.17$ |
| 0.99 | 1.24 | $60.16 \pm 2.69$ | $62.78 \pm 2.78$ | $2.62 \pm 3.80$ | $0.03 \pm 0.05$ | $-3.22 \pm 4.77$ |

(a) Rejected subset accuracy

(b) Overall test set accuracy

Figure 1: Test set accuracy for MNIST Digits using GWIN transformation for the given certainty threshold $\tau$. Figure 1a shows *BNN* and *BNN+GWIN* accuracy on the rejected subset for the Improved BNN. *% Reject* represents the percent of the 10,000 observations rejected by the classifier for the current certainty threshold. Figure 1b shows the accuracy of the *BNN* and *BNN+GWIN* on the entire test set for the Improved BNN. All results are presented as the mean over 10 runs and error bars show standard deviation.

Table 6: Test set accuracy for MNIST fashion on rejected observations using GWIN transformation for the given certainty threshold $\tau$. *BNN* and *BNN+GWIN* denote accuracy for the rejected subset using only the Improved BNN and the Improved BNN with GWIN reformulation, respectively. With no rejections ($\tau = 0$), the Improved BNN had an accuracy of 90.5%. *Overall Acc.* $\Delta$ denotes the change in accuracy while *% Error* $\Delta$ denotes the percent change in error rate for the entire subset when the GWIN is applied to rejected queries. All results are presented as the mean over 10 runs.

| $\tau$ | % Reject | IBNN Acc. | IBNN+GWIN Acc. | Rejected Acc. $\Delta$ | Overall Acc. $\Delta$ | % Error $\Delta$ |
|---|---|---|---|---|---|---|
| 0.50 | 0.19 | $36.35 \pm 9.30$ | $45.77 \pm 9.17$ | $9.42 \pm 11.05$ | $0.02 \pm 0.02$ | $-0.17 \pm 0.21$ |
| 0.70 | 2.52 | $44.78 \pm 2.87$ | $55.72 \pm 2.46$ | $10.95 \pm 2.35$ | $0.28 \pm 0.06$ | $-2.89 \pm 0.61$ |
| 0.80 | 4.02 | $47.11 \pm 2.37$ | $56.78 \pm 1.50$ | $9.67 \pm 2.93$ | $0.39 \pm 0.12$ | $-4.05 \pm 1.18$ |
| 0.90 | 6.13 | $49.62 \pm 1.35$ | $58.15 \pm 1.30$ | $8.53 \pm 1.91$ | $0.52 \pm 0.12$ | $-5.48 \pm 1.25$ |
| 0.95 | 8.19 | $52.62 \pm 2.15$ | $58.77 \pm 1.03$ | $6.15 \pm 2.11$ | $0.50 \pm 0.17$ | $-5.28 \pm 1.71$ |
| 0.99 | 12.37 | $57.18 \pm 1.11$ | $60.26 \pm 1.08$ | $3.09 \pm 1.59$ | $0.38 \pm 0.20$ | $-4.00 \pm 2.02$ |

(a) Rejected subset accuracy

(b) Overall test set accuracy

Figure 2: Test set accuracy for MNIST Fashion using GWIN transformation for the given certainty threshold $\tau$. Figure 2a shows *BNN* and *BNN+GWIN* accuracy on the rejected subset for the Improved BNN. *% Reject* represents the percent of the 10,000 observations rejected by the classifier for the current certainty threshold. Figure 2b shows the accuracy of the *BNN* and *BNN+GWIN* on the entire test set for the Improved BNN. All results are presented as the mean over 10 runs and error bars show standard deviation.

(a) MNIST Digits

(b) MNIST Fashion

Figure 3: Change in rejected sample certainty of the ground truth label for varying certainty rejection thresholds $\tau$ for the Improved BNN. Outliers are those values that fall outside of 1.5IQR and are denoted with diamonds.

## S3 GWIN Transformation Cost

For MNIST experiments using the LeNet-5 baseline, TensorFlow reports that a forward pass through the BNN requires 15,431,592 FLOPS and a forward pass through the WGWIN-GP generator requires 54,179,350 FLOPS. The additional cost of the rejection loop, which includes transforming the query and relabeling it, is then ~69.61 million FLOPS. The NVIDIA Titan X (Pascal) is rated at 11.0 TFLOPS, so the latency of rejection is ~0.06961 milliseconds on our devices.

Similarly, a forward pass through the Improved BNN baseline requires 61,829,923 FLOPS. The same GWIN architecture is used for both baselines, so the additional cost of the rejection loop is then ~116.0 million FLOPS, adding a latency of ~0.1160 milliseconds on our devices.

Note that the latency incurred by the classifier is dependent upon the classifier's architecture and that this latency would increase as the number of samples, and thus forward passes, increases. In general, the rejection and transformation will incur the cost of classification plus ~0.0542 milliseconds.

## S4 Bayesian Neural Network and Rejection Function Interaction

The Generative Well-intentioned framework does not make any strong assumptions about how the classifier and rejection function interact. As long as these two components support the interface described in Figure 4, they can be used with a GWIN.

The LeNet-5 Bayesian Neural Network and the Improved Bayesian Neural Network, detailed in Section S2, interact with the thresholded rejection function in the same way. We use Monte Carlo sampling to determine the BNN's predicted class and uncertainty metric. We first sample the model ten times for the given input $x_i$, effectively ensembling ten different networks. We treat the argmax of the mean logits as the class prediction $y'_i$. We treat the median of the probabilities for this predicted class as the certainty metric $c_i$. These two metrics are passed to the rejection function. We did not see a significant difference in WGWIN-GP performance when treating the mean as the certainty metric. Alternative approaches may consider the variance in the predicted class across models. Multiple passes through an approximation of a Bayesian network [3] or ensembling [4] have been used in related work to generate such uncertainties.

Figure 4: The expected interface of the rejection-based classifier. Aside from requiring the model to emit a certainty metric $c_i$ and label $y'_i$, no strong assumptions are made about the classifier. Since the classifier is fixed during generative training, it need not be a perceptron-based model. The rejection function $r : \{(c, y')\} \rightarrow \{\text{reject}, y'\}$ determines if the given observation is rejected or labeled.