[Reviews · NeurIPS 2019]

Reviewer 1



The paper is pleasant to read and presents a compelling innovation which complements GANs. The insight is simple yet powerful with strong potential impact in production systems relying on a quantification of uncertainty and offering the option to reject inputs. Strengths: 1) The exposition is flawless as is the experimental evaluation; 2) The method is original and simple and yet powerful as shown in the experimental section; 3) The method differs significantly from Defense GAN in a practically critical aspect: it only requires transformation of rejected examples. Weaknesses: 1) It may be interesting to comment on the added latency when re-processing an input as this may be a major practical problem when using such an approach in practice. For instance, what would be the added computational cost of the transformer in typical cases?

Reviewer 2



The method, using GANs to map low confidence examples to high confidence examples of the same class is highly original and shows promise as an effective method. The authors also provide a good description of how their work differs from Defense-GAN, and also notes some of the prior work on classifiers with a reject option (although they should also cite some of the more recent work doing so with DNNs e.g. [1]). The paper is clearly written, however there seems to be some information gaps with regards to network architecture and the mechanism for conditioning on the low confidence image x. Additionally there are several methods existing in the literature for conditioning the discriminator on label information: [1], [2], and concatenation and it's not clear from the paper which is used. The main drawback of this paper is that the datasets used (MNIST and FashionMNIST) are too toy to allow the reader to draw informed conclusions. The performance of the baseline classifier (from which the rejected images are derived) is extremely poor. There are simple convolutional network architectures that achieve >99.5% accuracy on MNIST compared to the 98% baseline and >95% on FashionMNIST compared to the 87.4% of the baseline used. This makes it difficult to evaluate how much of the improvements shown would be dwarfed by employing a better architecture. It's also not clear here what how the use of a Bayesian Neural Network interacts with the rejection function. How much different are the rejections if only a single model's confidence score is used for the rejection criteria rather than the median, what about the mean? Update: The authors have made effort to address my main concern of using a stronger baseline classifier, and training on harder datasets such as CIFAR10. They have also clarified some of the GAN architectural details. I still think that the incremental benefit of using an ensemble for computing rejection thresholds is at best dubious and doesn't obviously incorporate any kind of Bayesian model uncertainty and would like to see this clarified or investigated in the text. However due to the improvements, I will raise my score to a 6. [1] Geifman, Yonatan, and Ran El-Yaniv. "Selective classification for deep neural networks." Advances in neural information processing systems. 2017. [2] Odena, Augustus, Christopher Olah, and Jonathon Shlens. "Conditional image synthesis with auxiliary classifier gans." Proceedings of the 34th International Conference on Machine Learning-Volume 70. JMLR. org, 2017. [3] Miyato, Takeru, and Masanori Koyama. "cGANs with projection discriminator." arXiv preprint arXiv:1802.05637 (2018).

Reviewer 3



The author(s) developed and elegant architecture to employ GANs to improve the accuracy of the classifiers. They authors have also proposed a baseline implementation for the proposed GWIN architecture. This work can potentially have significant impact in areas in which the uncertain classification and mis-classification Classification of uncertain observations with increased certainty is a potentially important and useful problem. However, it is also at the risk of increasing the false positive rate. The authors fail to discuss such vital consequences and their experiments focus only on accuracy. For observations that cannot be classified with high certainty, it is somewhat unreasonable to translate them just for better recognizability under the specified classifier. In order to demonstrate the effectiveness of the proposed procedure, the experiments compare the prediction accuracy on rejected samples, since the pre-trained classifier is not expected to have high accuracy on samples where it does not have the confidence to judge. Also, when comparing with the baseline BNN, the BNN+GWIN has much more coefficients and has additional confident data set training steps. It is unclear whether the higher accuracy is a result of a more complicated model and larger training epochs. Last but not least, for image classification, uncertainty is often induced by noise in observations. So how would the proposed procedure compare with state-of-the-art denoising methods for overall prediction accuracy? On the other hand, if denoising can largely reduce the number of uncertain observations, then the contribution of this work is incremental at most.

[Author Response · NeurIPS 2019]

We thank the reviewers for acknowledging our contributions and for providing valuable feedback.

**To R#1: Transformer latency:** For MNIST experiments, TensorFlow reports that the LeNet-5 BNN requires 10,906,677 FLOPS and the WGWIN-GP generator requires 9,292,938 FLOPS. The additional cost of the rejection loop is then ~20.2 million FLOPS. The NVIDIA Titan X (Pascal) is rated at 11.0 TFLOPS, so the latency of rejection is ~0.02 milliseconds on our devices.

**To R#3: Additional WGWIN-GP architecture and concatenation details:** The architecture of the WGWIN-GP generator closely follows that described in the WGAN-GP paper, adding conditioning via the concatenation method referenced in Section 2.1. The conditioning image is flattened and concatenated with a noise vector and passed to the standard WGAN-GP generator as input. The critic architecture also follows the WGAN-GP paper. We concatenate the corresponding one-hot class label to the generated image and, similarly to [*1], to the output of each hidden convolution layer. We clarified the concatenation process in our paper and have added the missing hidden-layer citation. We also added detailed architecture diagrams for each network to our appendix. Thank you for pointing this out.

**To R#3: Bayesian neural network and rejection function interaction:** We use Monte Carlo sampling to determine the BNN's predicted class and uncertainty metric. We first sample the model ten times for the given input $x_i$, effectively ensembling ten different networks. We calculate the mean of the given class probabilities and treat the argmax as the class prediction $y_i'$. We treat the median of the probabilities for this predicted class as the certainty metric $c_i$. These two metrics are passed to the rejection function. We did not see a significant difference in WGWIN-GP performance when treating the mean as the certainty metric. Alternative approaches may consider the variance in the predicted class across models. Since uncertainty is represented by the difference in prediction across models, a single network prediction does not provide uncertainty information and thus should not be used alone.

**To R#3: Baseline models:** We reevaluate the WGWIN-GP using a stronger baseline (expanded LeNet-5 with batchnorm, dropout, and more convolution layers/filters; the exact architecture will be included in the paper) while the rejection function and WGWIN-GP are unchanged. The GWIN continues to have a positive impact. Results are shown in Table 1. CIFAR10 experiments are in progress.

**To R#4: Denoising methodologies:** A key distinction between our work and the papers mentioned by R#4 is that those papers and their related works focus on using GANs to increase the robustness of a classifier *during training* by generating out-of-distribution training data (similar to hallucination methods in the few-shot learning domain) while our method assumes a *fixed, pretrained* classifier and uses generative methods to translate novel, out-of-distribution examples to the confident distribution *during inference*. Since the GWIN framework learns representations that the classifier labels correctly with high confidence, these generative denoising methods can easily be paired with our framework: a classifier is trained using the aforementioned techniques and the GWIN is then used to transform any novel examples that the new classifier is not entirely robust to. Similarly to DefenseGAN, the flexibility and additive nature of our framework means that we can easily build atop these existing denoising methodologies. Since noise only represents a subset of out-of-distribution observations, we cannot rely entirely on denoising techniques to address classifier robustness. GWINs take a step towards a generalizable, principled framework for "rethinking" uncertain examples and leveraging classifier uncertainty. For completeness, we will add a more detailed Related Works section to the appendix describing how these methodologies address out-of-distribution robustness during the training process.

**To R#4: False positive rate:** Fig.5 shows the change in rejected sample certainty of the ground truth label for varying rejection thresholds. The WGWIN-GP increases confidence in the correct class more often than not. Depending on the classifier's accuracy on the rejected subset, the threshold can be tuned to maximize the expected number of correct classifications. Typically, a classifier is required to predict so higher accuracy implies fewer misclassifications and therefore fewer false positives. If true rejection is allowed, a model can still use the GWIN to perform a transformation and reject the original observation if the transformed observation is below some arbitrarily high threshold.

Table 1: Test set accuracy for Digits (top) and Fashion (bottom) on rejected observations using GWIN transformation for the given certainty threshold $\tau$ averaged over 10 runs. For $\tau = 0$, acc. is 99.2% (Digits) and 91.0% (Fashion).

| $\tau$ | % Reject | BNN Acc. | BNN+GWIN Acc. | Rejected Acc. $\Delta$ | Overall Acc. $\Delta$ | % Error $\Delta$ |
|---|---|---|---|---|---|---|
| 0.70 | 0.25 | $46.44 \pm 12.30$ | $59.96 \pm 7.28$ | $13.53 \pm 15.54$ | $0.03 \pm 0.04$ | $-3.42 \pm 4.15$ |
| 0.80 | 0.41 | $51.79 \pm 7.93$ | $66.13 \pm 8.39$ | $14.34 \pm 11.08$ | $0.06 \pm 0.05$ | $-6.39 \pm 4.90$ |
| 0.95 | 0.81 | $53.44 \pm 3.91$ | $68.60 \pm 4.50$ | $15.16 \pm 4.85$ | $0.12 \pm 0.04$ | $-12.64 \pm 3.66$ |
| 0.99 | 1.26 | $60.11 \pm 4.53$ | $67.63 \pm 3.62$ | $7.52 \pm 3.07$ | $0.10 \pm 0.04$ | $-9.77 \pm 3.69$ |
| 0.70 | 2.54 | $45.24 \pm 2.20$ | $57.10 \pm 2.75$ | $11.85 \pm 3.26$ | $0.30 \pm 0.09$ | $-3.17 \pm 0.93$ |
| 0.80 | 4.04 | $46.49 \pm 2.16$ | $57.49 \pm 2.67$ | $11.00 \pm 2.60$ | $0.45 \pm 0.11$ | $-4.64 \pm 1.09$ |
| 0.95 | 8.18 | $52.23 \pm 1.44$ | $59.18 \pm 1.31$ | $6.95 \pm 2.15$ | $0.57 \pm 0.19$ | $-5.99 \pm 1.93$ |
| 0.99 | 12.33 | $57.04 \pm 1.22$ | $61.61 \pm 0.86$ | $4.56 \pm 1.23$ | $0.56 \pm 0.15$ | $-5.88 \pm 1.48$ |

[*1] S. Reed, Z. Akata, X. Yan, L. Logeswaran, B. Schiele, H. Lee, arXiv preprint arXiv:1605.05396, 2016.


[Meta-Review · NeurIPS 2019]

The paper presents an approach of increasing classification accuracy by transforming the low confident predictions to the space of the inputs that are easier to classify. Well-written paper contrasting with related work, though missing some of the other relevant work. The limitations of the work is the simplicity of the datasets used. Would have liked to see more exposition on the out-of-distributions that are being modeled (and why de-noising methods won't work), especially since the experiments are performed on MNIST.